# Combining growth-promoting genes leads to positive epistasis in *Arabidopsis thaliana*

Hannes Vanhaeren[1,2][†], Nathalie Gonzalez[1,2][†], Frederik Coppens[1,2], Liesbeth De Milde[1,2], Twiggy Van Daele[1,2], Mattias Vermeersch[1,2], Nubia B Eloy[1,2], Veronique Storme[1,2], Dirk Inzé[1,2]*

[1]Department of Plant Systems Biology, Vlaams Instituut voor Biotechnologie, Ghent, Belgium; [2]Department of Plant Biotechnology and Bioinformatics, Ghent University, Ghent, Belgium

**Abstract** Several genes positively influence final leaf size in *Arabidopsis* when mutated or overexpressed. The connections between these growth regulators are still poorly understood although such knowledge would further contribute to understand the processes driving leaf growth. In this study, we performed a combinatorial screen with 13 transgenic *Arabidopsis* lines with an increased leaf size. We found that from 61 analyzed combinations, 39% showed an additional increase in leaf size and most resulted from a positive epistasis on growth. Similar to what is found in other organisms in which such an epistasis assay was performed, only few genes were highly connected in synergistic combinations as we observed a positive epistasis in the majority of the combinations with *samba*, *BRI1*[OE] or *SAUR19*[OE]. Furthermore, positive epistasis was found with combinations of genes with a similar mode of action, but also with genes which affect distinct processes, such as cell proliferation and cell expansion.

*For correspondence: diinz@psb.ugent.be

[†]These authors contributed equally to this work

**Competing interests:** The authors declare that no competing interests exist.

**Reviewing editor**: Daniel J Kliebenstein, University of California, Davis, United States

## Introduction

Since Bateson introduced the term epistasis to describe the phenomenon that some mutations seemed to be 'stopping' or 'standing above' the effect of other mutations (*Bateson, 1909*), it became clear that interactions between multiple genes influence many traits. Epistasis, or interaction between genes, therefore corresponds to any deviation from the expected phenotype, predicted by combining the effects of individual alleles or mutations (*Fisher, 1918*; *Phillips, 2008*). Only by identifying and understanding the nature of these underlying gene interactions, we will gain better insights in the regulation of complex traits and be able to dissect the architecture of biological networks.

In the last decade, numerous studies on the effect of pairwise gene perturbations have been conducted, primarily in the budding yeast *Saccharomyces cerevisiae*, to systematically evaluate epistasis for several characteristics, such as fitness or synthetic lethality (*Tong et al., 2004*; *Jasnos and Korona, 2007*; *St Onge et al., 2007*; *Dixon et al., 2009*; *Costanzo et al., 2010*, *2011*). These genome-scale genetic interactions studies were facilitated by the availability of large collections of deletion strains and the development of automated platforms to analyze the phenotypes of double mutants (*Scherens and Goffeau, 2004*). Since the first large-scale genetic interaction study in yeast identified 4000 genetic interactions among 1000 genes when analyzing synthetic lethality in double deletion mutants (*Tong et al., 2004*), the field advanced considerably. Currently, about 170,000 interactions are known among 5.4 million gene pairs screened to affect fitness (*Baryshnikova et al., 2010*; *Costanzo et al., 2010*). Interestingly, these studies have shown that the majority of the genes are

**eLife digest** Different individuals of the same species are not usually identical. Humans, for example, have a range of different values for various traits, such as height and weight, and similar variations are seen in other animals and plants. Some of this variation can be explained by the individual animals or plants inheriting different versions of the same gene. Moreover, two or more genes can sometimes interact to produce a bigger (or smaller) effect on given trait than would be expected from combining the effects of each gene. This phenomenon is called epistasis.

Leaf size in plants is a trait that is controlled by genes and by the environment. Genes can exert an effect by influencing how often the cells in the leaves divide and by influencing their expansion. In *Arabidopsis thaliana*, a small plant that has been widely studied by plant biologists, numerous genes influence leaf size. Now, in order to explore epistasis in plants, Vanhaeren, Gonzalez et al. have crossbred 13 *Arabidopsis* mutants that had leaves that were larger than those found in wild-type plants. The resulting offspring each inherited a different pair of mutant genes: one gene via the male pollen, and the other via the female egg cell. About 39% of the offspring plants had leaves that were even bigger than those of its parents. Moreover, in most cases the increase in leaf size was larger than expected.

Vanhaeren, Gonzalez et al. found that three of the 13 genes were responsible for the biggest increases in leaf size. Also, some of the biggest increases were caused by plants inheriting two genes that both caused the cells to divide more. Other large increases in leaf size were caused by interactions between a gene affecting cell division and a gene affecting cell expansion. By uncovering combinations of plant genes that increase leaf size, it might be possible to develop agricultural crops with enhanced yields.

infrequently connected in the genetic interaction network, while a small fraction of genes shows many interactions (*Baryshnikova et al., 2010*; *Costanzo et al., 2010*, *2011*).

In higher organisms, large collections of mutants often do not exist and/or the generation of double mutants is much more labor-intensive and time-consuming. However, in the nematode *Caenorhabditis elegans*, in *Drosophila* cell cultures and in human cell lines, global analysis of genetic interactions have been performed by making use of RNA interference libraries to generate double mutants (*Lehner et al., 2006*; *Byrne et al., 2007*; *Barbie et al., 2009*; *Horn et al., 2011*). In *C. elegans*, systematic mapping of interactions between genes functioning in the signaling and the transcriptional networks that regulate development also revealed high connectivity of a small proportion of genes in the network, while most genes have few interactions (*Lehner et al., 2006*). In plants, although large collections of mutants are available for some species such as *Arabidopsis* (*Alonso et al., 2003*; http://www.arabidopsis.org/), large-scale epistasis studies on double mutants are experimentally and practically virtually impossible to achieve. On a smaller scale, newly identified mutants in *Arabidopsis* are crossed with known mutants with similar phenotypes or within the same biological process to test for allelic interaction or epistasis. For example, genetic interactions among late flowering *Arabidopsis* mutants have been studied by generating double mutants (*Koornneef et al., 1998*). Further, genetic modifier screens are performed frequently through a random mutagenesis of individuals harboring one mutant gene to screen for second-site mutations that either enhance or suppress the primary phenotype. An example in relation to leaf size is the identification of enhancer mutations of *da1-1* further increasing leaf and seed size (*Li et al., 2008*; *Yao et al., 2008*; *Xu and Li, 2011*; *Fang et al., 2012*). While epistasis can easily be detected for qualitative traits, such as synthetic lethality, which are fairly straightforward to visually inspect, genetic interactions from quantitative traits, such as organ growth or gene expression, are more difficult to identify, especially in multicellular organisms (*Kroymann and Mitchell-Olds, 2005*; *Malmberg et al., 2005*; *Xu and Jia, 2007*; *Chapman et al., 2012*; *Steinhoff et al., 2012*; *Huang et al., 2014*). Estimating epistasis for quantitative categories of phenotypes implies calculating how much the phenotype of a double mutant deviates from an expected additive value based on the effect of the single mutations (*Fisher, 1918*), therefore requiring accurate measurements of the phenotype of the single and double mutants. Although enabling the identification of subtle interactions, these quantitative analyses of gene interactions are not easily amenable to large-scale studies of complex traits.

One example of such a complex quantitative trait in higher plants is leaf growth. Leaves are essential to capture solar radiation and convert it into chemical energy by photosynthesis, therefore contributing to a large part of plant biomass production. As for most plant organs, their determinate growth pattern results in a relatively constant size within a fixed environment. Leaf growth is mediated by a cell proliferation phase followed by a cell expansion phase that initiates at the leaf top and proceeds basipetally (*Donnelly et al., 1999*; *Andriankaja et al., 2012*). At least five different parameters contribute to the final leaf size (*Gonzalez et al., 2012*): the number of cells incorporated in leaf primordia; the rate of cell division; the developmental window of cell proliferation; the timing of meristemoid division; and the extent of cell expansion. Several genes have been described to, when downregulated or ectopically (over)expressed, increase the final leaf size in *Arabidopsis* (*Gonzalez et al., 2009*; *Krizek, 2009*; *Breuninger and Lenhard, 2010*) by affecting one or more processes governing leaf growth. Whereas much research has been done on single genes affecting leaf size, the interactions between these growth regulators remain unexplored. So far, only one case of positive epistasis in Arabidopsis leaf growth has been described when a dominant-negative point mutation in *DA1*, encoding an ubiquitin receptor, is combined with the knock-out of the *ENHANCER OF DA1* (*EOD)1/BIG BROTHER*, coding for an E3 ubiquitin ligase (*Li et al., 2008*).

In this study, we performed a combinatorial screen of transgenic *Arabidopsis* plants producing larger leaves to identify positive epistatic effects on leaf growth. We aimed to gain further insight in the links between genes controlling growth and the mechanisms driving leaf development. We obtained binary combinations by crossing 13 transgenic lines with an increased leaf size and measured leaf and rosette area of the single and double transgenics. We found that the leaf area of 38% of all combinations was larger than the sum of those of the single mutants, resulting in positive epistatic effects, whereas 23% of the combinations were smaller, showing a negative epistatic effect.

## Results

### Gene selection and experimental setup

To identify positive epistatic effects on leaf growth, we analyzed pairwise perturbations of 13 genes positively affecting final leaf size in a gain- or loss-of-function situation (*Table 1*) by measuring the individual and total leaf area. We used lines in the Col-0 background, homozygous for a single-locus insertion of the transgene of interest and shown to have a positive effect on all rosette leaves or a subset of those (*Cho and Cosgrove, 2000*; *Gonzalez et al., 2010*; *Spartz et al., 2012*). This enhanced leaf growth can result from the perturbation of genes affecting cell division and/or cell expansion. The downregulation of *SAMBA* disturbs the early stage of leaf development, since larger meristems are formed resulting in larger leaves containing more cells (*Eloy et al., 2012*). A point mutation in *DA1* or the downregulation of its enhancer, *EOD1*, leads to the production of larger leaves with more cells due to an extended cell proliferation phase (*Li et al., 2008*). Similarly, in plants overexpressing *ANGUSTIFOLIA3* (*AN3*), *AINTEGUMENTA* (*ANT*), *ARABIDOPSIS VACUOLAR-PYROPHOSPHATASE* (*AVP1*), *GROWTH-REGULATING FACTOR5* (*GRF5*) under the control of the constitutive 35S promoter or *BRASSINOSTEROID INSENSITIVE 1* (*BRI1*) under the control of its own promoter, larger leaves containing more cells are formed because of an extension of the cell proliferation phase (*Wang et al., 2001*; *Horiguchi et al., 2005*; *Li et al., 2005*). On the other hand, an increased cell proliferation at the edge of the leaf and a prolonged period of meristemoid division are observed when the miRNA *JAW* is overexpressed and the *PEAPOD* (*PPD*) genes are downregulated (*Palatnik et al., 2003*; *White, 2006*). When *GIBBERELLIN 20-OXIDASE 1* (*GA20OX1*) is overexpressed, an increase in cell number and cell size leads to the formation of larger leaves (*Huang et al., 1998*; *Gonzalez et al., 2010*). Finally, in plants overexpressing *EXPANSIN 10* (*EXP10*) and *SMALL AUXIN UP-REGULATED RNA 19* (*SAUR19*) fused to a GFP tag, bigger leaves containing larger cells are produced (*Cho and Cosgrove, 2000*; *Spartz et al., 2012*). Several of these leaf growth-promoting genes are involved in hormonal pathways, confirming the importance of plant hormones in the regulation of growth processes: *BRI1* encodes a brassinosteroid receptor, GA20OX1 catalyzes rate-limiting steps in late gibberellic acid (GA) biosynthesis, ANT has been suggested to be involved in auxin signal transduction and both AVP1 and SAUR19 in auxin transport (*Huang et al., 1998*; *Mizukami and Fischer, 2000*; *Wang et al., 2001*; *Li et al., 2005*; *Spartz et al., 2012*). To obtain pairwise perturbations, our strategy was to cross the

**Table 1.** Growth regulators and transgenics used for the binary combinations

| Gene name | Gene symbol | Gene ID | Line name | Perturbation | Cellular process promoted | Reference |
|---|---|---|---|---|---|---|
| ANGUSTIFOLIA3 | AN3 | AT5G28640 | AN3[OE] | OE | Cell division | (Horiguchi et al., 2005) |
| AINTEGUMENTA | ANT | AT4G37750 | ANT[OE] | OE | Cell division | (Mizukami and Fischer, 2000) |
| ARABIDOPSIS V-PYROPHOSPHATASE | AVP1 | AT1G15690 | AVP1[OE] | OE | Cell division | (Li et al., 2005) |
| BRASSINOSTEROID INSENSITIVE 1 | BRI1 | AT4G39400 | BRI1[OE] | OE | Cell division | (Wang et al., 2001; Gonzalez et al., 2010) |
| DA1 | DA1 | AT1G19270 | da1-1 | LOF | Cell division | (Li et al., 2008) |
| ENHANCER OF DA1-1/BIG BROTHER | EOD/BB | AT3G63530 | eod1-2 | LOF | Cell division | (Li et al., 2008) |
| EXPANSIN 10 | EXP10 | AT1G26770 | EXP10[OE] | OE | Cell expansion | (Cho and Cosgrove, 2000) |
| GIBBERELLIN 20 OXIDASE 1 | GA20OX1 | AT4G25420 | GA20OX1[OE] | OE | Cell division and expansion | (Huang et al., 1998; Gonzalez et al., 2010) |
| GROWTH REGULATING FACTOR5 | GRF5 | AT3G13960 | GRF5[OE] | OE | Cell division | (Horiguchi et al., 2005) |
| miR-JAW/ miRNA 319 | miR-JAW | AT4G23713 | jaw-D | OE | Cell division | (Palatnik et al., 2003) |
| PEAPOD | PPD | AT4G14713 and AT4G14720 | ami-ppd | LOF | Meristemoid division | (White, 2006; Gonzalez et al., 2010) |
| SAMBA | SAMBA | AT1G32310 | samba | LOF | Cell division and expansion | (Eloy et al., 2012) |
| SMALL AUXIN UP RNA 19 | SAUR19 | AT5G18010 | SAUR19[OE] | OE | Cell expansion | (Spartz et al., 2012) |

OE: over-expression, LOF: loss of function.

homozygous transgenic lines and to analyze the heterozygous progeny. We produced 102 heterozygous combinations, consisting of 78 paired combinations and 24 back-crosses with the wild type (WT) used as controls (*Figure 1—figure supplement 1*). Because the homozygous line can be used as pollen donor or receptor, care was taken that the crosses with the wild-type plants, producing the heterozygous control line, maintained the same directionality. For example, a cross between *ami-ppd* (♀) and *SAUR19^OE* (♂) was compared to the offspring of the crosses *ami-ppd* (♀) X WT (♂) and WT (♀) X *SAUR19^OE* (♂). This approach standardizes for possible maternal effects (*Scott et al., 1998*). Next, we checked the expression levels of the transgenes in the obtained heterozygous double mutants as well as in the heterozygous control lines. In the majority of the combinations, transgene expression levels were comparable with those of the heterozygous controls (*Figure 1—figure supplement 2*). In total, 61 combinations were used for further growth analysis. Sixteen plants per genotype were grown in three independent repeats and at 21 days after stratification (DAS), the size of each individual leaf of the rosette was measured, resulting in 56,505 data-points, enabling us to estimate potential gene interactions for these quantitative traits (*Figure 1—figure supplement 3*). Leaf area (LA) of the paired combinations was compared to a theoretical, expected if non-interacting value (EXPni), based on the size of the WT and both heterozygous controls. To estimate the EXPni, we applied an additive model on a multiplicative scale by transforming the data on log2 scale (*Koornneef et al., 1998*; *Phillips, 2008*; *Horn et al., 2011*):

$$log_2(LA_{EXPni}) = log_2\left(LA_{heterozygous\ control\ 1}\right) + log_2\left(LA_{heterozygous\ control\ 2}\right) - log_2\left(LA_{wild\ type}\right)$$

In order to identify combinations with synergistic or negative effects on leaf growth, we searched for significant leaf–genotype interactions (FDR <0.05). The significance of the difference between the EXPni and the observed value was determined using a mixed model ('Materials and methods'). This calculation and comparison was done for each combination (*Figure 1—figure supplement 4–64*). The LAs were analyzed using repeated measurements to take into account dependencies between the different leaves of the rosette. We also calculated the total rosette area, defined as the sum of all individual leaves. Similarly as for leaf area, a rosette EXPni was calculated.

## Identification of positive and negative epistasis effects on leaf growth

Among the 61 combinations analyzed, 23 pairwise crosses, almost 38%, were found to have a rosette size significantly exceeding the EXPni value (FDR <0.05, *Figures 1 and 2*). In the strongest synergistic combinations, such as *BRI1^OE-eod1-2*, *BRI1^OE-EXP10^OE*, *BRI1^OE-SAUR19^OE*, *GRF5^OE-SAUR19^OE*, *BRI1^OE-da1-1*, *ami-ppd-SAUR19^OE* and *samba-eod1-2* (at least 20% larger than the EXPni), the positive effect on size was observed for all rosette leaves. Remarkably, although out of the 13 genes that were selected for this screen only two are involved in increasing cell size (*EXP10^OE* and *SAUR19^OE*), almost half of the synergistic combinations arose from combining cell proliferation-stimulating gene perturbations with these two cell expansion-promoting genes, particularly with *SAUR19^OE* (*Figure 2*; *Table 1*). We also observed a positive epistasis in the majority of the combinations with *samba*, *BRI1^OE* or *SAUR19^OE*, suggesting that these growth regulators are more prone to lead to synergistic effects in binary combinations (*Figure 2—figure supplement 1A*).

Of all binary crosses analyzed, 39.2% resulted in plants with a rosette size exceeding that of both heterozygous control lines and the WT (*Figure 1*; *Supplementary file 1*). Interestingly, 16 combinations resulted from a synergistic effect, while eight were the result of an additive effect. Among the largest plants, synergistic (*GRF5^OE-SAUR19^OE* and *ANT^OE-SAUR19^OE*, 39% and 38% larger than the WT, respectively) and additive (*da1-1-GA20ox1^OE* and *ANT^OE-AVP1^OE*, 38% and 36% larger than the WT, respectively) effects could be found.

In addition, we also found that 23% of the combinations led to the formation of smaller rosettes than expected. We observed that mainly combinations with *jaw-D* and *ami-ppd* led to cases of negative epistasis. The total rosette area of these combinations was similar or much smaller than that of WT plants, such as *GRF5-jaw-D* (46% smaller than the WT), with the exception of *da1-1-ami-ppd*, which was larger than the WT, but smaller than *da1-1-Col-0*. (*Figure 1*).

In conclusion, from this screen, we found that more than one third of the combinations showed positive epistasis on leaf growth, resulting from combining either two genes both stimulating cell proliferation, or either one gene enhancing cell proliferation and the other cell expansion.

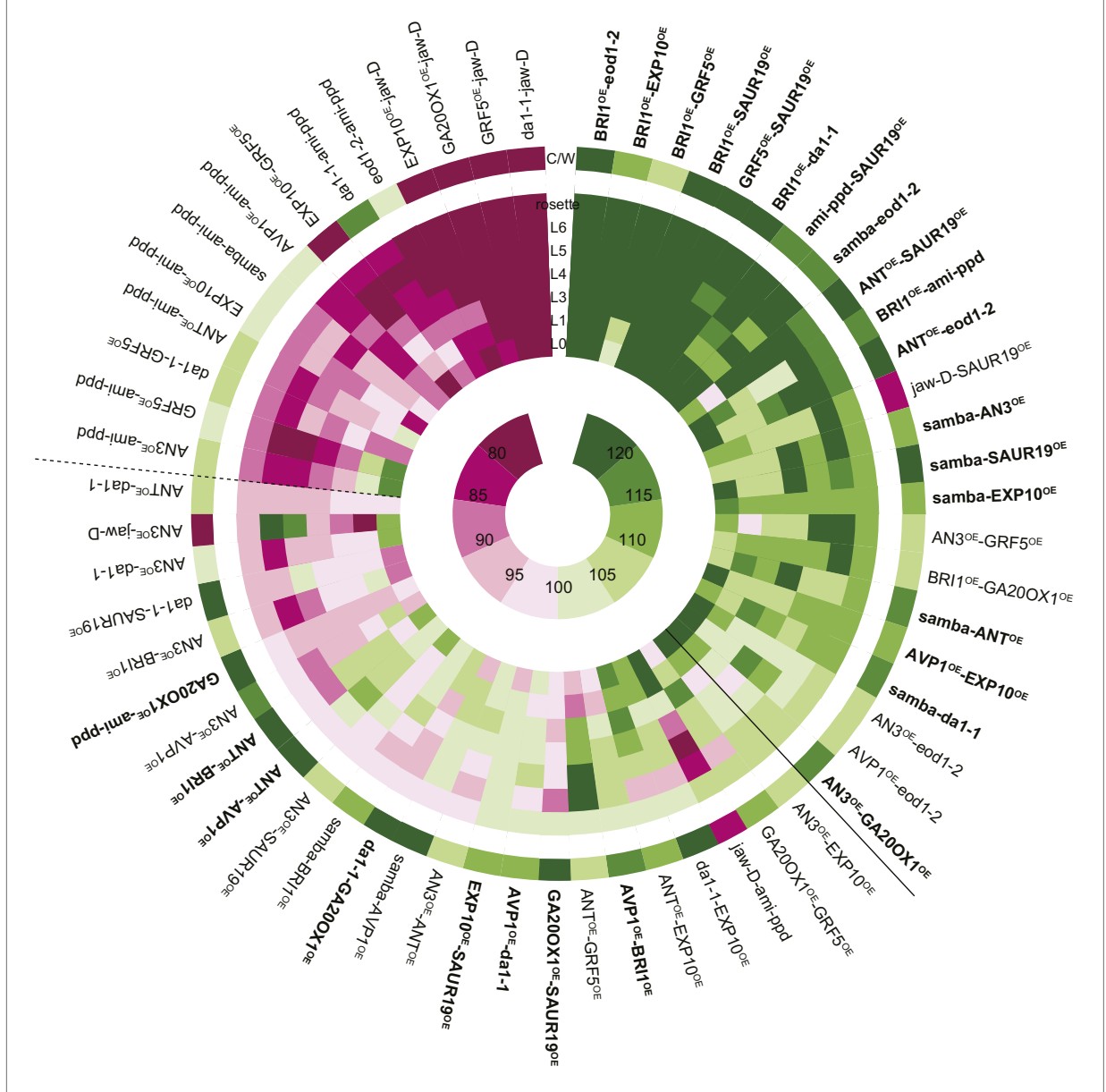

**Figure 1**. Heat map representing the effect of the binary combinations for rosette and leaf area. The outer ring shows the percentage of the rosette size of the combinations compared to the WT (C/W). In the middle rings, percentages of the observed sizes of the cotyledons (L0) until leaf 6 (L6) and the complete rosette are shown compared to the expected if non-interacting value (EXPni). Significant differences to the rosette EXPni value (FDR <0.05) allowed identifying synergistic interactions (black line) and negative interactions (dashed line) between two transgenic lines. The inner circle shows the color code with dark pink being the lowest and deep green being the highest value. Combinations that are at least 5% larger than each of their heterozygous controls are marked in bold.

The following figure supplements are available for figure 1:

**Figure supplement 1**. Overview of all heterozygous and homozygous combinations and their controls (Col x mutant or mutant x Col) obtained by crosses.

**Figure supplement 2**. Relative gene expression levels in the heterozygous binary combinations and their controls.

**Figure supplement 3**. Phenotypic analysis workflow.

*Figure 1. Continued on next page*

*Figure 1. Continued*

**Figure supplement 4**. Statistical output of the phenotypic data for the heterozygous combination AN3OE-ANTOE.

**Figure supplement 5**. Statistical output of the phenotypic data for the heterozygous combination *AN3OE-AVP1OE*.

**Figure supplement 6**. Statistical output of the phenotypic data for the heterozygous combination *AN3OE-BRIOE*.

**Figure supplement 7**. Statistical output of the phenotypic data for the heterozygous combination *AN3OE-da1-1*.

**Figure supplement 8**. Statistical output of the phenotypic data for the heterozygous combination *AN3OE-eod1-2*.

**Figure supplement 9**. Statistical output of the phenotypic data for the heterozygous combination *AN3OE-EXP10OE*.

**Figure supplement 10**. Statistical output of the phenotypic data for the heterozygous combination *AN3OE-GA20OX1OE*.

**Figure supplement 11**. Statistical output of the phenotypic data for the heterozygous combination *AN3OE-GRF5OE*.

**Figure supplement 12**. Statistical output of the phenotypic data for the heterozygous combination *AN3OE-jaw-D*.

**Figure supplement 13**. Statistical output of the phenotypic data for the heterozygous combination *AN3OE-ami-ppd*.

**Figure supplement 14**. Statistical output of the phenotypic data for the heterozygous combination *AN3OE-SAUR19OE*.

**Figure supplement 15**. Statistical output of the phenotypic data for the heterozygous combination *ANTOE-AVP1OE*.

**Figure supplement 16**. Statistical output of the phenotypic data for the heterozygous combination *ANTOE-BRI1OE*.

**Figure supplement 17**. Statistical output of the phenotypic data for the heterozygous combination *ANTOE-da1-1*.

**Figure supplement 18**. Statistical output of the phenotypic data for the heterozygous combination *ANTOE-eod1-2*.

**Figure supplement 19**. Statistical output of the phenotypic data for the heterozygous combination *ANTOE-EXP10OE*.

**Figure supplement 20**. Statistical output of the phenotypic data for the heterozygous combination *ANTOE-GRF5OE*.

**Figure supplement 21**. Statistical output of the phenotypic data for the heterozygous combination *ANTOE-ami-ppd*.

**Figure supplement 22**. Statistical output of the phenotypic data for the heterozygous combination *ANTOE-SAUR19OE*.

**Figure supplement 23**. Statistical output of the phenotypic data for the heterozygous combination *AVP1OE-BRI1OE*.

**Figure supplement 24**. Statistical output of the phenotypic data for the heterozygous combination *AVP1OE-da1-1*.

**Figure supplement 25**. Statistical output of the phenotypic data for the heterozygous combination *AVP1OE-eod1-2*.

**Figure supplement 26**. Statistical output of the phenotypic data for the heterozygous combination *AVP1OE-EXP10OE*.

**Figure supplement 27**. Statistical output of the phenotypic data for the heterozygous combination *AVP1OE-ami-ppd*.

**Figure supplement 28**. Statistical output of the phenotypic data for the heterozygous combination *BRI1OE-da1-1*.

**Figure supplement 29**. Statistical output of the phenotypic data for the heterozygous combination *BRI1OE-eod1-2*.

*Figure 1. Continued on next page*

*Figure 1. Continued*

**Figure supplement 30**. Statistical output of the phenotypic data for the heterozygous combination $BRI1^{OE}$-$EXP10^{OE}$.

**Figure supplement 31**. Statistical output of the phenotypic data for the heterozygous combination $BRI1^{OE}$-$GA20OX1^{OE}$.

**Figure supplement 32**. Statistical output of the phenotypic data for the heterozygous combination $BRI1^{OE}$-$GRF5^{OE}$.

**Figure supplement 33**. Statistical output of the phenotypic data for the heterozygous combination $BRI1^{OE}$-ami-ppd.

**Figure supplement 34**. Statistical output of the phenotypic data for the heterozygous combination $BRI1^{OE}$-$SAUR19^{OE}$.

**Figure supplement 35**. Statistical output of the phenotypic data for the heterozygous combination $da1$-$1$-$EXP10^{OE}$.

**Figure supplement 36**. Statistical output of the phenotypic data for the heterozygous combination $da1$-$1$-$GA20OX1^{OE}$.

**Figure supplement 37**. Statistical output of the phenotypic data for the heterozygous combination $da1$-$1$-$GRF5^{OE}$.

**Figure supplement 38**. Statistical output of the phenotypic data for the heterozygous combination $da1$-$1$-jaw-D.

**Figure supplement 39**. Statistical output of the phenotypic data for the heterozygous combination $da1$-$1$-ami-ppd.

**Figure supplement 40**. Statistical output of the phenotypic data for the heterozygous combination $da1$-$1$-$SAUR19^{OE}$.

**Figure supplement 41**. Statistical output of the phenotypic data for the heterozygous combination $eod1$-$2$-ami-ppd.

**Figure supplement 42**. Statistical output of the phenotypic data for the heterozygous combination $EXP10^{OE}$-$GRF5^{OE}$.

**Figure supplement 43**. Statistical output of the phenotypic data for the heterozygous combination $EXP10^{OE}$-jaw-D.

**Figure supplement 44**. Statistical output of the phenotypic data for the heterozygous combination $EXP10^{OE}$-ami-ppd.

**Figure supplement 45**. Statistical output of the phenotypic data for the heterozygous combination $EXP10^{OE}$-$SAUR19^{OE}$.

**Figure supplement 46**. Statistical output of the phenotypic data for the heterozygous combination $GA20OX1^{OE}$-$GRF5^{OE}$.

**Figure supplement 47**. Statistical output of the phenotypic data for the heterozygous combination $GA20OX1^{OE}$-jaw-D.

**Figure supplement 48**. Statistical output of the phenotypic data for the heterozygous combination $GA20OX1^{OE}$-ami-ppd.

**Figure supplement 49**. Statistical output of the phenotypic data for the heterozygous combination $GA20OX1^{OE}$-$SAUR19^{OE}$.

**Figure supplement 50**. Statistical output of the phenotypic data for the heterozygous combination $GRF5^{OE}$-jaw-D.

**Figure supplement 51**. Statistical output of the phenotypic data for the heterozygous combination $GRF5^{OE}$-ami-ppd.

**Figure supplement 52**. Statistical output of the phenotypic data for the heterozygous combination $GRF5^{OE}$-$SAUR19^{OE}$.

**Figure supplement 53**. Statistical output of the phenotypic data for the heterozygous combination jaw-D-ami-ppd.

**Figure supplement 54**. Statistical output of the phenotypic data for the heterozygous combination jaw-D -$SAUR19^{OE}$.

**Figure supplement 55**. Statistical output of the phenotypic data for the heterozygous combination ami-ppd -$SAUR19^{OE}$.

*Figure 1. Continued on next page*

*Figure 1. Continued*

**Figure supplement 56**. Statistical output of the phenotypic data for the heterozygous combination *samba*–AN3$^{OE}$.

**Figure supplement 57**. Statistical output of the phenotypic data for the heterozygous combination *samba* -ANT$^{OE}$.

**Figure supplement 58**. Statistical output of the phenotypic data for the heterozygous combination *samba* -AVP1$^{OE}$.

**Figure supplement 59**. Statistical output of the phenotypic data for the heterozygous combination *samba* -BRI$^{OE}$.

**Figure supplement 60**. Statistical output of the phenotypic data for the heterozygous combination *samba* -da1-1.

**Figure supplement 61**. Statistical output of the phenotypic data for the heterozygous combination *samba*-eod1-2.

**Figure supplement 62**. Statistical output of the phenotypic data for the heterozygous combination *samba* -EXP10$^{OE}$.

**Figure supplement 63**. Statistical output of the phenotypic data for the heterozygous combination *samba* -ami-ppd.

**Figure supplement 64**. Statistical output of the phenotypic data for the heterozygous combination *samba*-SAUR19$^{OE}$.

## Reciprocal and homozygous combinations

To strengthen the observed effects of pairwise perturbations and to further exclude that the observed phenotypes were influenced by maternal effects, we made reciprocal crosses of selected synergistic combinations (*SAUR19$^{OE}$-ami-ppd*, *EXP10$^{OE}$-BRI1$^{OE}$*, *SAUR19$^{OE}$-BRI1$^{OE}$*). We measured the

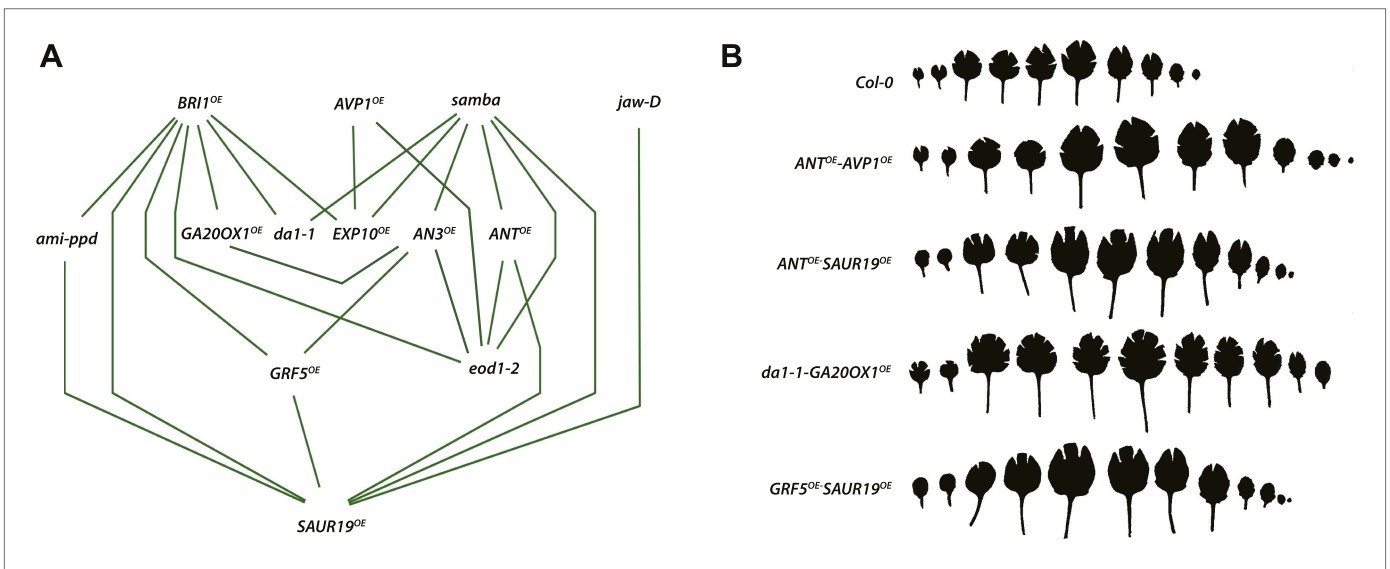

**Figure 2**. Network representing the combinations showing positive epistasis on total rosette area and leaf series of gene combinations with a large effect on leaf size. (**A**) The connections between two transgenics indicate the observation of a synergistic effect on rosette size. Two transgenics producing larger leaves resulting from an increased cell area are *SAUR19$^{OE}$* and *EXP10$^{OE}$*. (**B**) Both synergistic (*GRF5$^{OE}$-SAUR19$^{OE}$* and *ANT$^{OE}$-SAUR19$^{OE}$*) and additive combinations (*da1-1-GA20ox1$^{OE}$* and *ANT$^{OE}$-AVP1$^{OE}$*) lead to plants strongly enlarged up to 39% compared to the WT. In order to flatten the leaves for area measurements, cuts were made in the blade.

The following figure supplements are available for figure 2:

**Figure supplement 1**. Occurrence of the growth-regulating genes in a (**A**) synergistic combination and (**B**) negative combinations.

**Figure supplement 2**. Phenotype of the homozygous combination *da1-1-SAUR19$^{OE}$*.

leaf area at 21 DAS and could confirm the synergistic effects for all three combinations (*Figure 3A*, *Figure 3—figure supplement 1–3*). Next, we generated homozygous lines for two synergistic combinations, *ami-ppd-SAUR19^OE* and *samba-eod1-2*, and one additive combination, producing nevertheless a very large rosette, *da1-1-SAUR19^OE*. Transgene expression levels in these homozygous lines were verified and found comparable to those in the homozygous single lines (*Figure 3—figure supplement 4*). We confirmed a synergistic effect on the rosette sizes in homozygous *ami-ppd-SAUR19^OE* and *samba-eod1-2* plants (24% and 8% larger than the rosette EXPni respectively) (*Figure 3B*, *Figure 3—figure supplement 5,6*). The combination *da1-1-SAUR19^OE*, which produced among the largest plants in the screen, but did not enhance leaf size synergistically, was also found to be particularly large when homozygous, since its rosette size was 61% larger than that of the WT (*Figure 3B*, *Figure 3—figure supplement 7*, *Figure 2—figure supplement 2*). From these experiments we could confirm the observed positive epistatic effects in a selected set of double mutants from our screen of heterozygous combinations in a reciprocal direction and/or homozygous status.

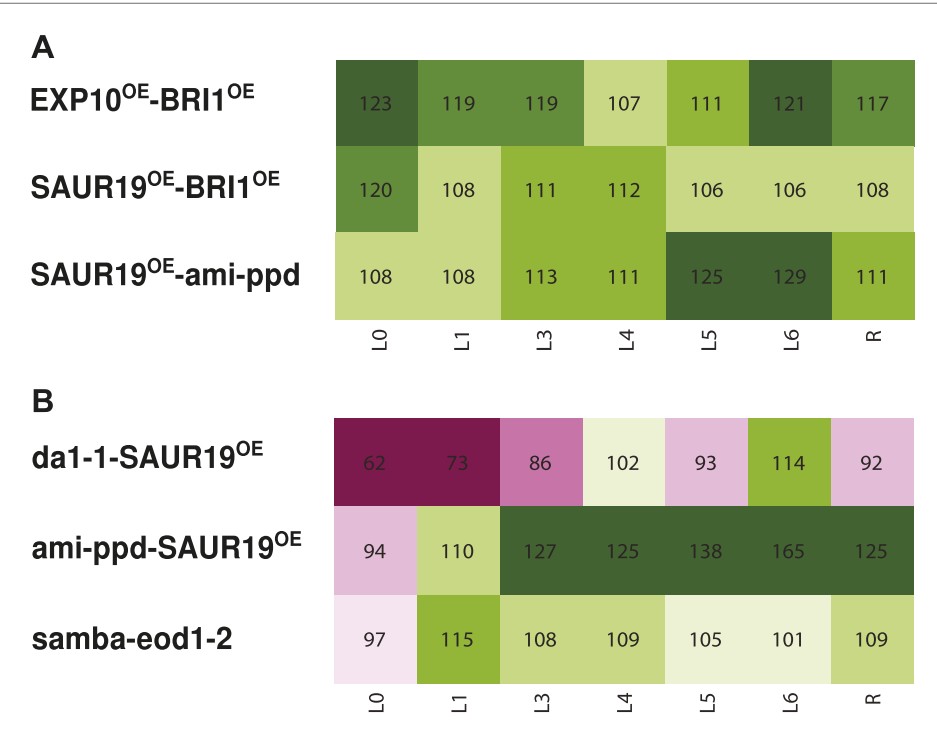

**Figure 3**. Heat map representing the effect of the binary combinations for rosette and leaf area (**A**) in reciprocal heterozygous crosses and (**B**) homozygous lines. C/W represents the percentage of the rosette size of the combinations compared to the WT. Percentages of the observed sizes of the cotyledons (L0) until leaf 6 (L6) and the complete rosette are shown compared to the expected if non-interacting value (EXPni). The color code represents the range of differences with dark pink being the lowest and deep green being the highest value.

The following figure supplements are available for figure 3:

**Figure supplement 1**. Statistical output of the phenotypic data for the heterozygous combination *EXP10^OE-BRI1^OE*.

**Figure supplement 2**. Statistical output of the phenotypic data for the heterozygous combination *SAUR19^OE-BRI1^OE*.

**Figure supplement 3**. Statistical output of the phenotypic data for the heterozygous combination *SAUR19^OE-ami-ppd*.

*Figure 3. Continued*

**Figure supplement 4**. Relative gene expression levels in the homozygous binary combinations and their controls.

**Figure supplement 5**. Statistical output of the phenotypic data for the homozygous combination *ami-ppd-SAUR19OE*.

**Figure supplement 6**. Statistical output of the phenotypic data for the homozygous combination *samba*-eod1-2.

**Figure supplement 7**. Statistical output of the phenotypic data for the homozygous combination da1-1-*SAUR19OE*.

## Cellular analysis of *ami-ppd-SAUR19OE*

In order to explain the cause for the observed synergistic phenotype at a cellular level, we quantified cell numbers and cell size in the homozygous combination *ami-ppd-SAUR19OE*. In the *ami-ppd* line, in which *PPD1* and *PPD2* expression is downregulated, the increased leaf size results from a prolonged division of meristemoids (**White, 2006**), whereas overexpression of *SAUR19* leads to cell enlargement (**Spartz et al., 2012**). Samples of leaf 3 were harvested at 21 DAS, cleared and cell drawings of the abaxial epidermis were analyzed. As shown in **Figure 4**, the larger leaves of *SAUR19OE* contain less but larger cells, whereas in leaves of *ami-ppd* more cells are produced. In the latter, an observed reduction in average cell area results from the presence of a larger amount of smaller cells surrounding the

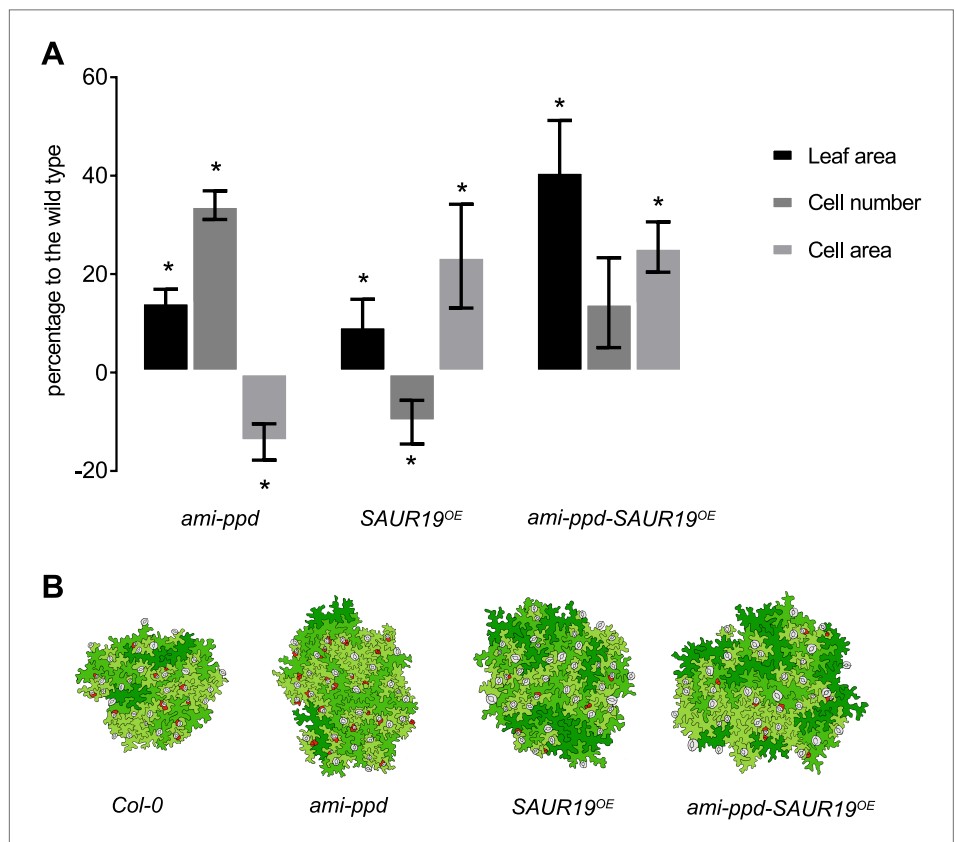

**Figure 4**. Cellular basis of the difference in leaf size observed for the homozygous line *amippd-SAUR19OE* and the corresponding controls. (**A**) The graphs represent the percentage difference of leaf area, cell number and cell area between a transgenic and the WT. (n = 3; *p<0.05). (**B**) Representative drawing of cells in the different lines. Cells are colored in function of their area. Red: cells smaller than 1.25 E$^{-4}$ mm$^2$, light green: cell area ranging from 1.25 E$^{-4}$ mm$^2$ to 1.6 E$^{-3}$ mm$^2$, medium green: cell area ranging from 1.6 E$^{-3}$ mm$^2$ to 3.2 E$^{-3}$ mm$^2$, dark green: cells larger than 6.4 E$^{-3}$ mm$^2$, stomata are marked in grey.

stomata which do not reach the mature wild-type size (*Figure 4B*). In the homozygous *ami-ppd-SAUR19^OE* line, we observed an increased cell number compared to the WT, but to a lower extend than in the *ami-ppd* line, and an increased cell area similar to that of *SAUR19^OE*. Thus the effect of *SAUR19^OE* allows for an increased cell expansion of the many small cells resulting from *PPD* downregulation.

## Discussion

In order to identify potential interactions existing within the genetic network regulating leaf growth, we pairwised combined 13 gene perturbations each leading to an enhanced leaf size and looked for positive interactions resulting in an increased leaf area larger than the additive combination of the single perturbations.

From this screen, we found that 61% of the paired perturbations showed epistasis: 38% of the studied gene combinations further enhanced leaf organ size synergistically and 23% negatively influenced leaf size. Studies using limited numbers of mutations, random or affecting a specific trait, also showed that epistasis is common, although lower levels of interactions were found (*Clark and Wang, 1997*; *Magwire et al., 2010*). In *D. melanogaster*, for example, 35 of 128 (27%) of random paired mutations showed epistasis (*Clark and Wang, 1997*). Larger-scale studies, in systems allowing automated quantitative assays, identified between 13 and 35% of epistatic effects (*Byrne et al., 2007*; *St Onge et al., 2007*). The large number of interactions we identified could be explained by the fact that we studied a set of perturbations, including loss and gain of function, leading to one particular phenotype, namely an increase of leaf area. In model systems permitting genome-wide genetic interactions assays, all genes are either knocked down or knocked out and these perturbations can therefore affect the studied trait, for example fitness, by increasing it or decreasing it. In *D. melanogaster*, the study of ten mutations leading to an increased life span showed that paired combinations have high levels of connections, with 21 significant epistatic interactions in males and/or females (47%) observed (*Magwire et al., 2010*).

Interestingly, three genes, *SAMBA*, *BRI1* and *SAUR19*, were found to lead to a synergistic effect in the majority of combinations they were part of. Large-scale genetic interaction studies in yeast and nematodes have shown that most genes in a network have only a few interactions, while a limited number of genes show multiple interactions and are therefore considered as network hubs mediating across-process connections (*Lehner et al., 2006*; *Baryshnikova et al., 2010*; *Costanzo et al., 2010*). Despite the relative small scale of the study presented here, our observations suggest that *SAMBA*, *BRI1* and *SAUR19* play a central role in the leaf growth regulatory networks.

Two of these highly connected genes in synergistic combinations, *BRI1* and *SAUR19,* have a known role in hormone signaling. Interestingly, yeast studies have shown that highly connected genes in a genetic network tend to be pleiotropic and multi-functional (*Costanzo et al., 2010*), similar to plant hormones which regulate multiple processes. BRI1 is a receptor of the brassinosteroid (BR) hormone which plays a crucial role in several biological processes, including leaf growth, as severe dwarfism is observed in *bri1* mutants and other mutants of the BR biosynthesis and signaling pathways (*Clouse et al., 1996*; *Vert et al., 2008*). *BRI1* is highly expressed in all organs during early seedling development (*Friedrichsen et al., 2000*) similarly to highly connected genes in yeast which show high mRNA levels (*Costanzo et al., 2010*). Additionally, introduction of *BRI1^OE* into *P10-CKX3^OE*, which has a smaller rosette size than WT plants, results in positive epistatic effects on shoot growth (*Vercruyssen et al., 2011*), highlighting the importance of this gene in leaf growth regulation. *SAUR19* belongs to the family of *SAUR* genes known to be rapidly and strongly induced by auxin (*Hagen and Guilfoyle, 2002*), which plays a major role in the initiation of leaf primordia, the formation of vascular patterns and leaf shape, but also in the regulation of leaf cell expansion (*Chen et al., 2001*; *Wilmoth et al., 2005*; *Scarpella et al., 2010*). *SAUR19* is a positive regulator of cell expansion, most likely through the modulation of auxin transport (*Spartz et al., 2012*). Our findings therefore suggest that alterations of BR or auxin signaling in the binary combinations could potentiate the effect of several growth-promoting genes.

Interactions between BR and other plant hormones have been shown for several physiological and developmental processes (*Choudhary et al., 2012*; *Li and He, 2013*; *Zhu et al., 2013*). BR and auxin interactions exist at multiple levels, including hormone synthesis, transport, signal transduction, and gene transcription. For example, microarray studies have revealed similar effects of BR and auxin on a large number of genes, including a member of the SAUR family, *SAUR15* (*Goda et al., 2004*; *Nemhauser et al., 2004*; *Walcher and Nemhauser, 2012*). Interestingly, exogenous application of both hormones leads to a synergistic induction of many common targets (*Nemhauser et al.,*

2004; *Vert et al., 2008*). In addition, auxin can increase the biosynthesis of BRs (*Chung et al., 2011*; *Yoshimitsu et al., 2011*) and the BR-regulated BIN2 kinase contributes to a synergistic increase in auxin-induced gene expression (*Vert et al., 2008*). The overexpression of both *BRI1* and *SAUR19*, involved in BR perception and auxin transport, respectively, could therefore amplify the effect of both hormones, hereby leading to the observed synergism in leaf growth.

Studies in yeast have shown that most genetic interactions occur between genes involved in the same biological process, except for highly connected genes (*Tong et al., 2004*; *Costanzo et al., 2010*). In agreement with these studies, we found that by combining $AN3^{OE}$ with $GRF5^{OE}$, shown to interact in a yeast two-hybrid assay (*Horiguchi et al., 2005*), the leaf size is increased more than expected. A similar effect is seen when $BRI1^{OE}$ and *ami-ppd*, both producing enlarged and curled leaves (*Wang et al., 2001*; *White, 2006*), are combined. Moreover, *PPD* genes regulate the division of dispersed meristemoid cells in the leaf epidermis, which will give rise to the stomatal lineage (*White, 2006*) and BRs have been shown to control stomatal development (*Gudesblat et al., 2012*; *Kim et al., 2012*; *Khan et al., 2013*). In addition, in *BRI1* overexpressing seedlings, *PPD2* has been reported to be downregulated (*Gonzalez et al., 2010*). In literature, the combination of *da1-eod* has been reported to show a positive epistatic effect on leaf growth. Both proteins are suggested to work in ubiquitin-mediated proteolysis that could modulate the activity of a shared, yet unknown target (*Li et al., 2008*). However, not only combining growth-regulating genes that are interconnected can lead to larger phenotypes than expected, also combining cell proliferation with cell expansion leads to positive effects on leaf size as found in the combinations *ami-ppd-SAUR19*$^{OE}$, $GRF5^{OE}$-$SAUR19^{OE}$ and *samba-EXP10*$^{OE}$. In addition, the combination of lines positively affecting distinct growth processes seems to allow compensating negative effects sometimes observed when constitutively expressing or strongly downregulating growth regulators, such as observed in *ami-ppd-SAUR19*$^{OE}$ (*Figure 4*). In plants overexpressing *GRF5* and *jaw-D*, each promoting cell proliferation, a reduction in cell area has also been reported (*Gonzalez et al., 2010*). Interestingly, when these genes are combined with $SAUR19^{OE}$, a synergistic effect on growth can be observed, similar to *ami-ppd-SAUR19*$^{OE}$. This suggests that the double transgenic line can acquire the benefits from both genes and therefore enhance leaf size more than expected. Such compensation could be lacking in the negative combinations we observed, therefore leading to the formation of smaller plants than expected. For example, by combining *GRF5* and *jaw-D*, both producing more but smaller cells, the negative effect on leaf size could be caused by overstimulation of cell division that affects the overall growth as observed when *E2Fa* and *DPa* are overexpressed simultaneously (*De Veylder et al., 2002*). These findings highlight the challenge of studying genetic interactions in multicellular organisms, compared to single cell systems such as yeast. Genetic interactions observed at the organ level can reflect connections between genes working in the same pathway, but also the interconnection of several processes such as cell division and cell expansion which occur in different cell types and tissues, at different rates and developmental stages. Although yeast is heavily used as a model to identify genetic interactions, it will be essential to also use multicellular organisms as a model for genetic interactions to capture the complex relationship between developmental processes.

In this study we searched for binary combinations of growth-regulating genes exhibiting an increase in leaf growth larger than the addition of the two single transgenic parents. In plants and animals, the phenomenon of heterosis or hybrid vigor corresponds to the increased performance of a hybrid offspring compared to its parents (*Schnable and Springer, 2013*). Heterosis has been proposed to arise from various mechanisms such as intra-allelic dominance and intra-allelic over-dominance, but emerging evidence also exists for the contribution of inter-gene interactions, or epistasis (*Kaeppler, 2012*; *Chen, 2013*; *Schnable and Springer, 2013*). Our findings suggest that differences in expression of growth-promoting genes in natural variants could lead to synergistic effects in hybrids. For example, one could imagine that in one variant, *PPD* is lowly expressed, whereas in another variant *SAUR19* is highly expressed. The combination of both genes in a cross of natural variants could lead to a synergistic increase in leaf size as observed in our study. Heterosis could therefore originate, in part, from the assembly of the effects of various pairwised combinations of growth-regulating genes. Another theory to explain heterosis describes the fact that hybrid vigor allows for the compensation of small negative alleles (*Kaeppler, 2012*; *Chen, 2013*; *Schnable and Springer, 2013*). In our study, we also found that negative effects of some perturbations can be compensated in pairwised combinations, allowing the appearance of a synergistic effect on growth, such as in the cross *ami-ppd-SAUR19*$^{OE}$.

So far, genetic engineering of crops mainly has been commercially successful for input traits, such as insect tolerance and herbicide resistance (http://www.isaaa.org). Engineering quantitative,

yield-related traits, such as drought tolerance and enhanced biomass production, turned out to be much more difficult. The current study illustrates that gene combinations have great promise to successfully engineer quantitative traits. Furthermore, the observation that genes stimulating cell proliferation combine remarkably well with genes enhancing cell expansion, argues for a need for further in-depth analysis of how single genes promote organ growth. A better understanding of the mode of action of growth- and/or yield-enhancing genes will allow for rationalizing which gene stacks have the highest probability to give successful results. Future prospects of combining multiple genes or even entire circuits of networks using synthetic biology approaches offer great perspectives to further enhance crop yield and to deliver sufficient food for the growing world demand.

# Materials and methods

## Plant material

Seeds of *A. thaliana* (L) Heyhn. ecotype Columbia-0 (Col-0) and all mutants (*Table 1*) were grown on soil and kept in the same growth room for 25 days, when flower stalks started to emerge. For all single insertion locus transgenic lines, binary crosses were made in one direction; for a selection of these lines, reciprocal crosses were made and homozygous lines were produced.

## Growth analysis

All plants were grown on plates containing half-strength MS medium (*Murashige and Skoog, 1962*) supplemented with 1% sucrose with a density of one plant per 4 cm². The seeds were stratified for 2 days at 4°C and placed in growth rooms kept at 21°C and 16-hr day/8-hr night cycles. Plants were grown in three experiments, consisting of 16 replicates per experiment. To ensure environmental conditions are similar between the experiments, they were performed consecutively in the same growth chamber on the same shelf. To prevent positional effects on plant growth, all plates were randomized every 2 days. We set out to grow all genotypes simultaneously in three repeats, but due to germination issues with some seed batches, a total of 5 experiments have been performed to obtain three repeats for each genotype, with the exception of the cross $ANT^{OE}$-*da1-1* for which we could obtain results in one repeat. At 21 DAS, individual leaves (cotyledons and rosette leaves) were dissected at the base of the petiole and their area was measured with ImageJ v1.45 (NIH; http://rsb.info.nih.gov/ij/).

## RNA extraction, cDNA preparation and q-RT-PCR

Total RNA was extracted from flash-frozen seedlings with TRIzol reagent (Invitrogen, Belgium). To eliminate the residual genomic DNA present in the preparation, the RNA was treated by RQ1 RNAse-free DNase according to the manufacturer's instructions (Promega, The Netherlands, http://www.promega.com) and purified with the RNeasy Mini kit (Qiagen, The Netherlands, http://www.qiagen.com). Complementary DNA was made with the iScript cDNA Synthesis kit from Biorad (Biorad, Belgium, http://www.bio-rad.com) according to the manufacturer's instructions. Q-RT-PCR was done on a LightCycler 480 (Roche, Belgium, http://www.roche.com) in 384-well plates with LightCycler 480 SYBR Green I Master (Roche) according to the manufacturer's instructions. Primers were designed with the Primer3 (http://frodo.wi.mit.edu/) (*Supplementary file 2*). Data analysis was performed using the ΔΔCT method (*Pfaffl, 2001*), taking the primer efficiency into account. The data was normalized using six normalization genes (UBQ10, CDKA1, CBP20, AT1G13320, AT2G32170, and AT2G28390) according to the GeNorm algorithm (*Vandesompele et al., 2002*).

## Microscopy for epidermal cell size measurements

For the cellular analysis, samples of leaf 3 were cleared in 70% ethanol and mounted in lactic acid on a microscope slide. The total leaf blade area was measured for 10 representative leaves under a dark-field binocular microscope. Abaxial epidermal cells along the complete proximal–distal axis of the leaves were drawn with a microscope equipped with differential interference contrast optics (DM LB with 403 and 633 objectives; Leica) and a drawing tube. Photographs of leaves and scanned cell drawings were used to measure leaf and cell area, respectively, with ImageJ v1.45 (NIH; http://rsb.info.nih.gov/ij/), from which the cell numbers were calculated (*De Veylder et al., 2001*).

## Statistical analysis

The leaf series data analysis yields the size of each individual leaf of the rosette. From this data, the rosette area was calculated by summating the area of all separate leaves. A mixed model analysis was performed

on the $log_2$ transformed rosette areas using Genotype as a fixed factor. Each experiment was repeated three times. This Experiment effect was included as a random factor in the model to account for correlation between measurements done within the same experiment. The Genotype*Experiment interaction was included in the model when it was found to be significant ($p<0.05$), based on a likelihood ratio test. For all described combinations the variation attributed to the Genotype was much larger than that attributed to the interaction between Genotype and Experiment. Severe outliers, caused by germination problems, were removed prior to the analysis. Least square means estimates for the rosette area were calculated.

Significant differences between the rosette area (RA) of the cross and its parental lines, as well as with the reference plants, were determined using the described mixed model (WALD-type III tests of fixed effects). To test for synergistic effects following null hypothesis was set up:

$$log_2\left(\widehat{RA}_{cross}\right) = log_2\left(\widehat{RA}_{control1}\right) + log_2\left(RA_{control2}\right) - log_2\left(\widehat{RA}_{wildtype}\right)$$

Through log-transformation of the data, we apply an additive model with a multiplicative scale (*Koornneef et al., 1998*; *Phillips, 2008*; *Horn et al., 2011*). As control lines the appropriate heterozygous parental lines were used.

By rearranging terms we get:

$$log_2\left(\widehat{RA}_{cross}\right) - log_2\left(\widehat{RA}_{control1}\right) - log_2\left(RA_{control2}\right) + log_2\left(\widehat{RA}_{wildtype}\right) = 0$$

A FDR multiple testing correction was applied. Synergistic effects were assumed when the null hypothesis was rejected at a FDR level of 0.05. The model was fit with the mixed procedure from SAS. To estimate repeatability (broad sense heritabilities at the individual level), the mixed model was refit with genotype, experiment and genotype*experiment as random terms in the model (*Supplementary file 3*).

The leaf series data was analyzed using repeated measurements analysis with either the hpmixed or mixed procedure from SAS. Data for leaves up to leaf 6 was included in the analysis. The four variance-covariance structures available in the procedure were tested and the best structure was determined based on the AIC values. For all combinations the unstructured structure was selected as the best. The mean model included the main effects Genotype and Leaf, and their interaction term. To account for dependencies of observations made within the same experiment, experiment was added as random factor in the model. Based on a likelihood ratio test the Genotype*Experiment interaction was incorporated in the model when $p<0.05$. Several contrast hypotheses were set up. For all leaves, the area in the reference line was compared to that in the cross and both parental lines. Synergistic effects of the cross were determined for each leaf, as described previously.

For the cross $ANT^{OE}$-*da1-1*, there was only one experiment that yielded results, therefore Experiment was not included as a factor in the model.

All statistical analyses were performed with SAS 9.3 (SAS Institute Inc., 2011, Cary, North Carolina). Residual diagnostics were carefully examined.

## Acknowledgements
We thank all colleagues of the Systems Biology of Yield group for fruitful discussions and Annick Bleys for help in preparing the manuscript.

## Additional information

### Funding

| Funder | Grant reference number | Author |
| --- | --- | --- |
| Belgian State, Science Policy Office | Interuniversity Attraction Poles Programme IUAP P7/29 | Hannes Vanhaeren, Nathalie Gonzalez, Frederik Coppens, Liesbeth De Milde, Twiggy Van Daele, Mattias Vermeersch, Nubia B Eloy, Veronique Storme, Dirk Inzé |

| Funder | Grant reference number | Author |
|--------|------------------------|--------|
| Ghent University | Bijzonder Onderzoeksfonds Methusalem Project no. BOF08/01M00408 | Hannes Vanhaeren, Nathalie Gonzalez, Frederik Coppens, Liesbeth De Milde, Twiggy Van Daele, Mattias Vermeersch, Nubia B Eloy, Veronique Storme, Dirk Inzé |
| Agency for Innovation through Science and Technology | Predoctoral fellowship | Hannes Vanhaeren |
| Ghent University | Multidisciplinary Research Partnership, Biotechnology for a Sustainable Economy no. 01MRB510W | Hannes Vanhaeren, Nathalie Gonzalez, Frederik Coppens, Liesbeth De Milde, Twiggy Van Daele, Mattias Vermeersch, Nubia B Eloy, Veronique Storme, Dirk Inzé |

The funders had no role in study design, data collection and interpretation, or the decision to submit the work for publication.

## Author contributions

HV, NG, Conception and design, Acquisition of data, Analysis and interpretation of data, Drafting or revising the article; FC, LDM, TVD, MV, NBE, VS, Acquisition of data, Analysis and interpretation of data; DI, Conception and design, Drafting or revising the article

# Additional files

### Supplementary files

• Supplementary file 1. Percentage rosette area of a heterozygous combination compared to the heterozygous parents. Column 'line': heterozygous combination, Column '% to parA': if a combination A is combined with B, par A represents the parent A, Column '% to parB': if a combination A is combined with B, parB represents the parent B, Column 'Pcross-parA': pvalue, Column 'Pcross-parB': pvalue.

• Supplementary file 2. List of primers used for the Q-RT-PCR analysis.

• Supplementary file 3. Estimation of the interaction between genotype and experiment.

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
