## [Decision Letter]

Thank you for sending your work entitled “Better together: Combining plant growth-promoting genes leads to positive epistasis” for consideration at eLife. Your article has been favorably evaluated by Deputy Editor Detlef Weigel and 2 reviewers, one of whom is a member of our Board of Reviewing Editors.

The data provide an important beginning to the understanding of how genetic loci in plants may be stacked to control quantitative phenotypes. Additionally it connects this to a preliminary analysis of molecular connections. This will serve to help stimulate a new avenue of research in both basic and applied plant biology.

The Reviewing editor and the other reviewer discussed their comments before we reached this decision, and the Reviewing editor has assembled the following comments to help you prepare a revised submission.

1) The description of the experimental design needs to be clarified to allow the reader to understand how the data was improved. Be as explicit as possible.

2) Both reviewers were confused by what was and was not considered significant in Figure 1 as there was discussion of a linear model with FDR but then a threshold was utilized. This needs to be clarified.

3) There should be much better use of the existing yeast data on the same topic to help this paper in a compare and contrast fashion.

4) Commentary on both negative and positive epistasis should be included as both were observed.

---

## [Author Response]

*1) The description of the experimental design needs to be clarified to allow the reader to understand how the data was improved. Be as explicit as possible*.

In order to clarify the experimental design of the study presented in the manuscript, several additions to the text were made.

-We first added a supplemental figure (Figure 1—figure supplement 1) to describe how the 102 heterozygous combinations, consisting of 78 paired combinations and 24 back-crosses with the wild type (WT) used as controls were produced. In this figure, the reciprocal crosses and the double homozygous lines produced are also mentioned.

-We also clarify the sentence describing the use of a multiplicative model by explaining that the data was log transformed: “To estimate the EXPni, we applied an additive model on a multiplicative scale *by transforming the data on log2 scale*.”

-Finally, in the Material and methods section, we added the following text to explain in more detail how plants were grown: “Plants were grown in three experiments, consisting of 16 replicates per experiment. To ensure environmental conditions are similar between the experiments, they were performed consecutively in the same growth chamber on the same shelf. To prevent positional effects on plant growth, all plates were randomized every two days. We set out to grow all genotypes simultaneously in three repeats, but due to germination issues with some seed batches, a total of 5 experiments have been performed to obtain three repeats for each genotype, with exception of the cross *ANT*^*OE*^*-da1-1* for which we could obtain results in one repeat”.

*2) Both reviewers were confused by what was and was not considered significant in*
Figure 1
*as there was discussion of a linear model with FDR but then a threshold was utilized. This needs to be clarified*.

We realize that the explanation given in the manuscript about what is significant and what we consider as being significantly synergistic is confusing. To avoid any confusion, we now consider all combinations having a FDR<0.05 as showing epistasis.

Originally, we used a threshold of 10% because our main objective was to look for highly synergistic combinations based on size differences. All the combinations above this 10% threshold (20 in total) show a significant difference (FDR<0.05) compared to the expected calculated value for rosette area. However, 3 extra combinations also have a FDR<0.05 and are therefore significantly synergistic. This change leads to an increase in the percentage of synergistic combinations.

For clarity, we now adapted the text as follows:

-“In order to identify combinations with synergistic or negative effects on leaf growth, we searched for significant leaf-genotype interactions (FDR<0.05).”

-The percentage of synergistic combinations is therefore also changed: from 33 to 38. “Among the 61 combinations analyzed, 23 pairwise crosses, almost 38%, were found to have a rosette size significantly exceeding the EXPni value (FDR<0.05, (Figures 1 and 2).”

-In the Figure 1 we also indicate the significant combinations (FDR<0.05, positive (black line) and negative (dashed line)) to ease the read of the figure.

-For the confirmed synergistic effect in the homozygous combinations we have now written: “We confirmed a synergistic effect on the rosette sizes in homozygous *ami-ppd-SAUR19*^*OE*^ and *samba-eod1-2* plants (24% and 8% larger than the rosette EXPni respectively)” instead of “We confirmed a strong synergistic effect on the rosette sizes in homozygous *ami-ppd-SAUR19*^*OE*^ plants (24% larger than the rosette EXPni). Although in the homozygous *samba-eod1-2* plants, the rosettes were 8% larger than the EXPni, a strong synergistic effect was observed for the first leaves (15% larger than the EXPni).”

*3) There should be much better use of the existing yeast data on the same topic to help this paper in a compare and contrast fashion*.

In order to show how our data compare or are different from existing yeast data but also data from other organisms, we extended the discussion paragraph as follows:

-From this screen, we found that 61% of the paired perturbations show epistasis: 38% of the studied gene combinations further enhancing leaf organ size synergistically and 23% negatively influencing leaf size. Studies using limited numbers of mutations, random or affecting a specific trait, also showed that epistasis is common, although lower levels of interactions were found. In *D. melanogaster*, for example, 35 of 128 (27%) of random paired mutations showed epistasis. Larger-scale studies, in systems allowing automated quantitative assays, identified between 13 and 35 % of epistatic effects. The large number of interactions we identified could be explained by the fact that we studied a set of perturbations, including loss and gain of function, leading to one particular phenotype, namely an increase of leaf area. In model systems permitting genome-wide genetic interactions assays, all genes are either knocked down or knocked out and these perturbations can therefore affect the studied trait, for example fitness, by increasing it or decreasing it. In *D. melanogaster*, the study of ten mutations leading to an increased life span showed that paired combinations have high level of connections, with 21 significant epistatic interactions in males and/or females (47%) observed.

-*BRI1* is highly expressed in all organs during early seedling development similarly to highly connected genes in yeast, which show high mRNA levels.

These findings highlight the challenge of studying genetic interactions in multicellular organisms, compared to single cell systems such as yeast. Genetic interactions observed at the organ level can reflect connections between genes working in the same pathway, but also the interconnection of several processes such as cell division and cell expansion that occur in different cell types and tissues, at different rates and in developmental stages. Although yeast is heavily used as a model to identify genetic interactions, it will be essential to also use multicellular organisms as a model for genetic interactions to capture the complex relationship between developmental processes.

*4) Commentary on both negative and positive epistasis should be included as both were observed*.

As previously mentioned, our objective was mainly to look for combinations showing positive synergistic effects. However, we agree that the generated dataset contains more information that should be presented and discussed. Therefore we adapted the text to include negative epistasis as follows:

In the result section: “In addition, we also found that 23% of the combinations lead to the formation of smaller rosettes than expected. We observed that mainly combinations with *jaw-D* and *ami-ppd* lead to cases of negative epistasis. The total rosette area of these combinations was similar or much smaller than that of WT plants, such as *GRF5-jaw-D* (46% smaller than the WT), with the exception of *da1-1-ami-ppd*, which is larger than the WT, but smaller than *da1-1-col*. (Figure 1).”

In the Discussion section: “Such compensation could be lacking in the negative combinations we observed, therefore leading to the formation of smaller plants than expected. For example, by combining *GRF5* and *jaw-D*, both producing more but smaller cells, the negative effect on leaf size could be caused by overstimulation of cell division that affects the overall growth as observed when *E2Fa* and *DPa* are overexpressed simultaneously.